# In-plane Fermi surface mapping of ZrSiS and HfSiS by de Haas-van Alphen oscillations

**Bruno Gudac[1], Mihovil Bosnar[2], Filip Orbanić[1], Trpimir Ivšić[3], Ivan Kokanović[1] and Mario Novak[1]⋆**

**1** Department of Physics, Faculty of Science, University of Zagreb, 10000 Zagreb, Croatia
**2** Division of Theoretical Physics, Ruđer Bošković Institute, 10000 Zagreb, Croatia
**3** Institute of Solid State Physics, TU Wien, 1040 Vienna, Austria

⋆ mnovak@phy.hr

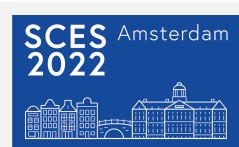 *International Conference on Strongly Correlated Electron Systems (SCES 2022)*
## Abstract

We have conducted a comprehensive experimental investigation of the in-plane de Haas van Alphen quantum oscillations in ZrSiS and HfSiS and compared the results to the ab-initio calculations. We found a moderately good agreement for a set of frequencies located between 100 T and 200 T. These could be associated with extremal orbitals of the electron pockets located around the $k_z = \pm\pi/2c$ planes at the edge of the 1st Brillouin zone. In contrast, the experimentally detected low-frequency set of oscillations found around 20 T could not be reproduced by the calculations. Further refinement of the ab-initio calculation has to be conducted to fully capture all the observed features of the Fermi surface in ZrSiS and HfSiS.

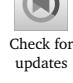

## 1 Introduction

Topological semimetals are a class of gapless systems with topologically protected crossings of the energy bands. When certain symmetries are present, energy bands will have different quantum numbers in such a way that they are unable to hybridize when crossing in energy occurs [1–3]. Topological semimetals host linearly dispersing electron excitations described by an effective massless/massive 3D Dirac equation. Systems with topologically nontrivial band-structure provide a possibility to explore the fundamental physics of relativistic particles in condensed matter, resulting in many exotic properties [4–11]. In general, linearity of energy dispersion in these systems ensures low effective masses and high mobilities [12].

ZrSiS and HfSiS belong to the topological nodal-line semi-metals group. The two compounds are iso-structural, crystallizing in a tetragonal square-net crystal structure with non-symmorphic P4/nmm (129) symmetry [13]. ZrSiS is one of the first reported topological

nodal-line semimetals and has been widely studied [14–17, 19, 20]. It contains two types of nodal lines. The first type corresponds a set of nodal lines close to the Fermi level forming a cage-like structure [19, 21]. They are not topologically protected since only C4v symmetry is present and are prone to a small gap opening induced by the spin-orbit interaction [20]. The second type of nodal line is topologically protected by nonsymmorphic symmetry and is located around 1eV below the Fermi level [19, 21]. ZrSiS hosts a plethora of interesting magnetoresistance phenomena magnetoresistance [22–24], unusual magnetic susceptibility [25], magnetic brake-down effects [17, 18, 26, 27].

In contrast, the physical properties and the Fermi surface shape of HfSiS have been much less explored. Since Hf is heavier than Zr, stronger spin-orbit coupling is present resulting in a bigger gap at the crossing of nodal lines close to the Fermi level which should influence the Fermi surface shape.

In this paper, we present de Haas-van Alphen (dHvA) quantum oscillations in ZrSiS and HfSiS single crystals. Angular dependence of the oscillation was measured for magnetic fields parallel to the [001] direction (to verify of the sample quality) and selected directions in the ab-plane. This measurements enabled us to better understand the shape of the Fermi surface in these two compounds. We have detected pronounced oscillations which, after the Fast-Fourier transformation (FFT), revealed well-defined frequency peaks for the in-plane directions. Using the ab-initio DFT calculations we obtained a set of frequencies which were then compared to the experimental results reveling the details the Fermi surface shape.

## 2 Experimental

ZrSiS and HfSiS single crystals were grown by the chemical vapor transport method using iodine as a transport agent. Zr and Hf sponges were alloyed with Si by arc-melting several times and ground in a mortar. Next, ZrSi and HfSi powders were mixed with sulfur with the addition of Si lost in the arc-melting process and vacuum sealed in an evacuated quartz ampule with a small amount of iodine (10mg/cm$^3$). The ampule was placed in a tube furnace in a temperature gradient of 950$^o$C-850$^o$C for two weeks. Single crystals of ZrSiS and HfSiS were extracted from a cold end of the ampule (see Fig.1d)).

The structure of the single crystals was analyzed by using x-ray powder and Laue diffraction, Fig.1 c) and Fig.1 e). ZrSiS and HfSiS crystallize in the tetragonal crystal structure within the space group P4/nmm [13]. The crystal structure consists of quintuple layers of S-Zr-Si-Zr-S, as shown in Fig.1 a). Si strongly bonds with four Zr in a tetrahedral position, while the layers are connected with a weak van der Waals type interaction through S atoms.

Weak interlayer bonding makes material susceptible to cleaving. Using a wire saw samples of an approximate mass of 25 mg were cut and used for the measurement. The magnetization was performed with a Quantum Design MPMS3 superconducting quantum interference vibrating sample magnetometer (SQUID-VSM) in a field up to 7T. The samples were mounted on a quartz rod with a minimal background contribution. Before proceeding with the in-plane measurements the samples were characterized in the [001] direction (c-axis). Isothermal out-of-plane (B||c) magnetization of ZrSiS and HfSiS single crystal is shown in Fig.1 f). Both samples exhibit dHvA oscillations superimposed on paramagnetic and diamagnetic backgrounds, respectively. ZrSiS in contrast to HfSiS shows strong Zeeman splitting and the data agree with work done by Hu et al. [16].

The ab-initio calculations to further analyze the experimental data were performed at the level of density functional theory (DFT). The DFT calculations were performed using Quantum ESPRESSO (QE) package. All calculations were performed with the PBEsol exchange-correlation functional and the corresponding Pseudo-Dojo norm-conserving pseudopotentials.

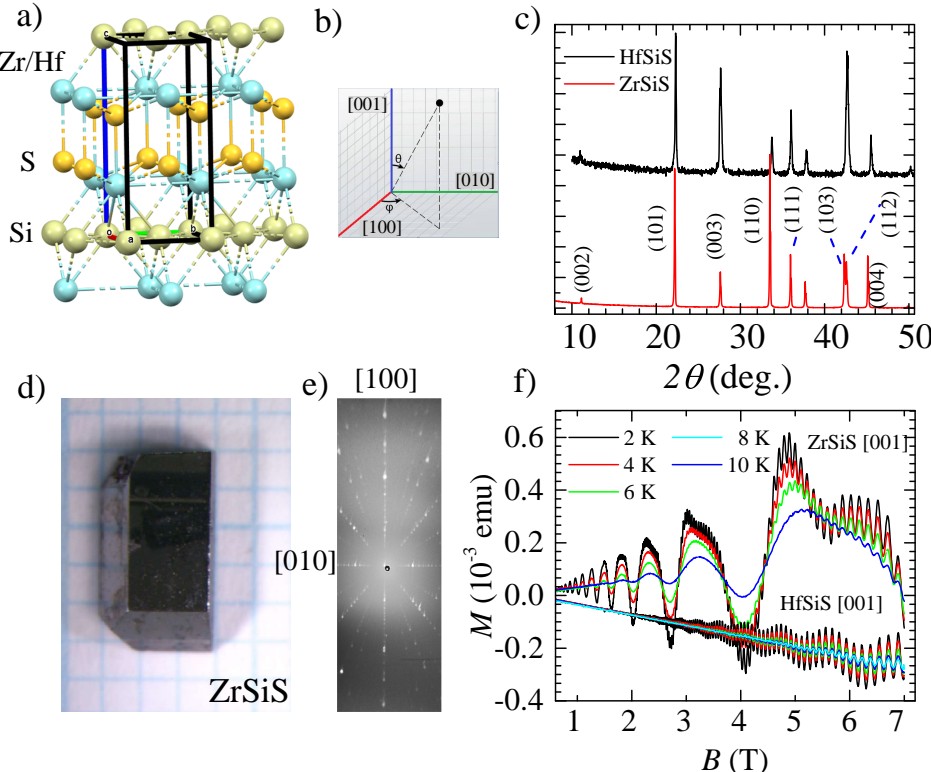

Figure 1: a) ZrSiS and HfSiS isostructural tetragonal unit cell. b) Depiction of magnetic field orientation. $\phi$ represents the in-plane magnetic field orientation. c) XRD pattern of ZrSiS (red) and HfSiS (black). Both compounds show equivalently positioned characteristic peaks. d) Single-crystal of ZrSiS could grow up to 60 mg and several mm in lateral dimensions. e) Laue pattern of ZrSiS single-crystal showing high-quality of the sample. f) Magnetisation measured by a SQUID magnetometer shows strongly pronounced dHvA oscillations superposed on a linear background. A magnetic field was orientated along the c-axes ([001] direction).

First, the lattice parameters of cells under consideration were optimized. For these optimizations, QE's built-in algorithm was used. The plane wave cutoff for optimizations was set to 1440 Ry and the I. Brillouin zone was sampled by 46x46x20 Monkhorst-Pack grid, while the electronic occupations were smeared using the Marzari-Vanderbilt scheme with parameter $\sigma = 0.01$ Ry. Both the shape and parameters of the cell were allowed to vary until the pressure on the cell decreased under 0.5 kbar, while the atoms in the cell moved until the force fell under 0.001 Ry/Bohr. This procedure resulted in the cell parameters listed in Table 1. To obtain the precise Fermi surfaces, a static SCF calculation was performed on an optimized structure with the plane wave cutoff reduced to 60 Ry and the number of points in the Monkhorst-Pack grid increased to 92x92x40. For this calculation, the spin-orbit coupling was also enabled. Fermi surfaces were calculated from the results of SCF using QE's postprocessing tool. The calculated Fermi surfaces were visualized using the FermiSurfer package. The frequencies of de Haas-van Alphen oscillations were found from the extracted Fermi surfaces using the Supercell K-space Extremal Area Finder (SKEAF) package [28].

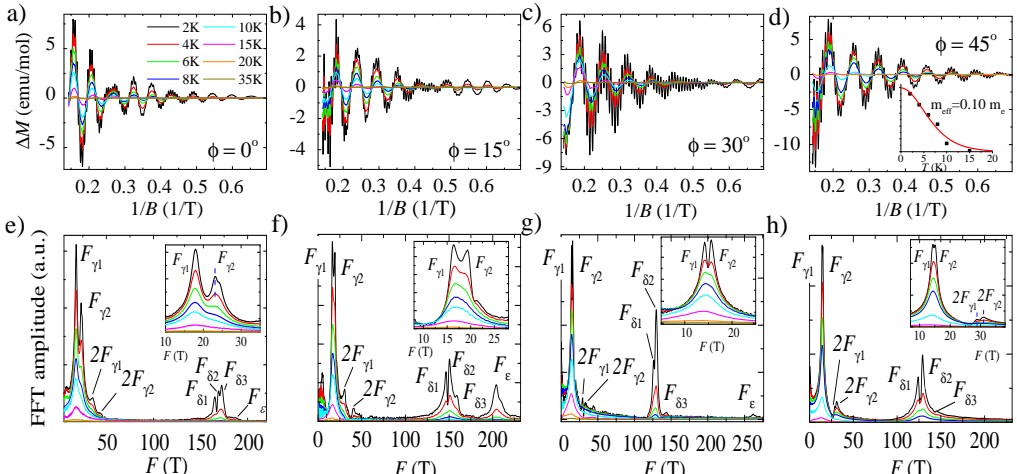

Figure 2: a)-d) Temperature dependence of the dHvA oscillations in ZrSiS for differ-ent in-plane orientations ($\phi = 0^0, 15^o, 30^o, 45^o$) of magnetic field. For all measured directions, oscillations are strongly pronounced and the multi-frequency nature of the oscillations can be observed. $\phi = 0^o$ corresponds to [100] or [010] and $\phi = 45^o$ to [110] direction. e)-h) Corresponding FFT analysis of the dHvA oscillations for different in-plane directions. Two distinct sets of frequencies can be observed. The low-frequent set consists of two frequencies $F_{\gamma1}$ and $F_{\gamma2}$ and its angular evolution can be tracked down. For $\phi = 45^o$ $F_{\gamma1} \approx F_{\gamma2}$ Insets: d) Effective mass for $F_{\gamma1}$ in the direction $\phi = 45^o$ using Lifshitz-Kosevich theory is $0.1(2)m_e$. e)-h) Zoom-in on the angular evolution of frequencies $F_{\gamma1}$ and $F_{\gamma2}$.

Table 1: Comparison of the experimental and DFT cell parameters for ZrSiS and HfSiS [13].

|          | a (Å) | c (Å) | a/c      |
|----------|-------|-------|----------|
| Exp ZrSiS | 3.544 | 8.055 | 0.4400   |
| DFT ZrSiS | 3.519 | 8.029 | 0.438286 |
| Exp HfSiS | 3.498 | 7.929 | 0.4410   |
| DFT HfSiS | 3.520 | 8.000 | 0.440    |

# 3   Results and discussion

Quantum-oscillation phenomena can be used as a technique for measuring the low-temperature Fermi surface morphology. They are described by the Lifshitz-Kosevich the-ory [29]. Oscillations of various physical quantities with magnetic field for 3D-electron gas can be described by:

$$\Delta X = A_0 A_T A_D A_S \left(\frac{B}{F}\right)^{1/2} \cos\left(2\pi\frac{F}{B} + \phi_M\right). \tag{1}$$

$\Delta X$ stands for oscillating physical quantity, magnetization in our case, $A_0$ is dimensionless constant, $F$ is the oscillation frequency and $\phi_M$ are phase factors that take into account the nature of the extremal orbit and the Berry phase. $A_T$, $A_D$, and $A_S$ are temperature, Dingle, and spin dimensionless pre-factors, respectively. $A_T = 2\pi^2(k_B T/\hbar\omega_c)/\sinh[2\pi^2(k_B T/\hbar\omega_c)]$, $A_D = \exp[2\pi^2(k_B T_D/\hbar\omega_c)]$, $A_S = \cos[\pi g m_e/2m_c]$, where $T_D$ is Dingle temperature, $\omega_c = eB/m_c$ is cyclotron mass, $m_e$ is the free electron mass and $m_c$ is the effective cyclotron

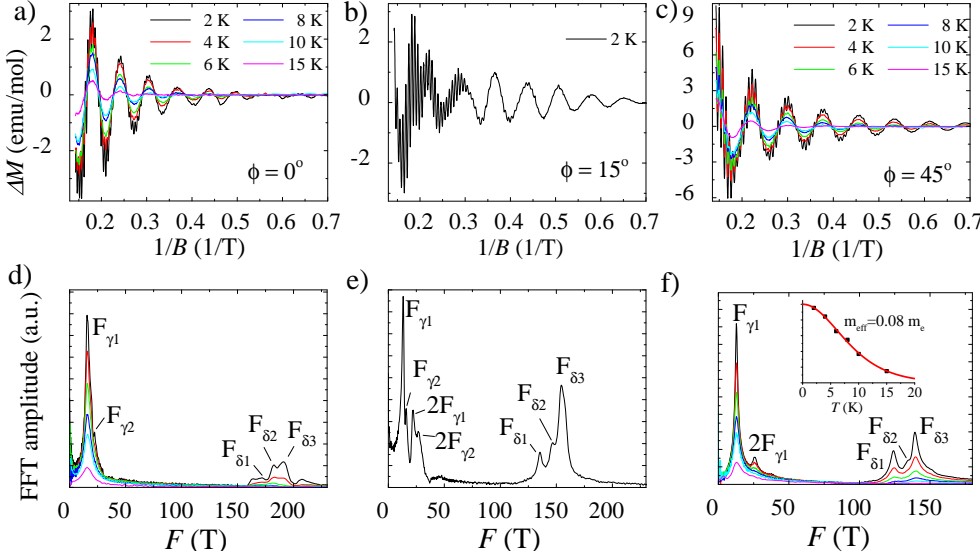

Figure 3: a)-c) HfSiS temperature dependant dHvA oscillations for three different azimuth directions. The oscillations are well pronounced and comparable in magnitude to ZrSiS oscillations. d)-f) Corresponding FFT analysis. As in the case of ZrSiS, we can observe two sets of frequencies. Inset: Effective mass for $F_{\gamma 1}$ in the direction $\phi = 45^o$ is $0.08(2)m_e$.

mass. Dingle temperature is further related to the quantum relaxation time $\tau_q = \hbar/2\pi k_B T_D$ from which one can estimate quantum mobility $\mu_q = e\tau/m_c$.

Figures Fig.2 a)-d) show in-plane dHvA oscillations in ZrSiS for four different azimuth directions, $\phi = 0^o, 15^o, 30^o, 45^o$, and eight different temperatures. In all directions, the oscillations are clearly visible and consist of several frequencies. Due to the high sample quality and low effective masses, the oscillations can be detected even at 1 T.[1] By applying the FFT analysis (see Fig.2 e)-h) ) we get information on frequency spectrum of the oscillations. The frequency spectrum can be divided into two groups; the low-frequency set of peaks $F_{\gamma 1}$ and $F_{\gamma 2}$, with its second harmonics $2F_{\gamma 1}$ and $2F_{\gamma 2}$, and the high-frequency set of peaks $F_{\delta 1}$, $F_{\delta 2}$, $F_{\delta 3}$ and $F_{\epsilon}$. In the low-frequency region, angular peak evolution can be traced by changing $\phi$. At $\phi = 0^o$ $F_{\gamma 1} = 17.8$ T and $F_{\gamma 2} = 23.2$ T are well separated, whereas at $\phi = 45^o$ $F_{\gamma 1} = 14.5$ T and $F_{\gamma 2} = 15.0$ T have (practically) merged into a single peak. This indicates that frequencies $\gamma 1$ and $\gamma 2$ originate from the same part of the Fermi surface. The high-frequency set of peaks also shows movement with $\phi$ which will later be compared to numerical DFT results. Estimating the effective mass by the Lifshitz-Kosevich theory gives, in the case of $\phi = 45^o$, $m_{eff} = 0.10(2)m_e$. The low effective mass is responsible for the long-living low-frequency oscillations which can be observed even at 20K.

HfSiS in Fig.3 a)-f) shows similar behavior of the low and high set of frequencies. The onset of oscillations is also positioned at around 1 T. The low-frequencies $F_{\gamma 1}$ and $F_{\gamma 2}$ show almost identical angular behavior as in the case of ZrSiS. At $\phi = 0^o$ the frequency peaks are separated, whereas at $\phi = 45^o$ they merge into a single peak. Low frequency effective mass is comparable to ZrSiS being $m_{eff} = 0.08(2)m_e$ for $\phi = 45^o$.

To get a better understanding of the origin of the observed dHvA oscillations we have performed DFT calculations. In Fig.4 a) and d) we show a direct comparison of experimental and calculated frequencies as a function of azimuth angle $\phi$. In the case of ZrSiS the frequencies $F_{\delta 1,2,3}$ (Fig.4 a)) show a satisfactory agreement with the calculated values. The $F_{\epsilon}$

---

[1]For ZrSiS in [001] direction oscillations can be detected even below 0.5 T.

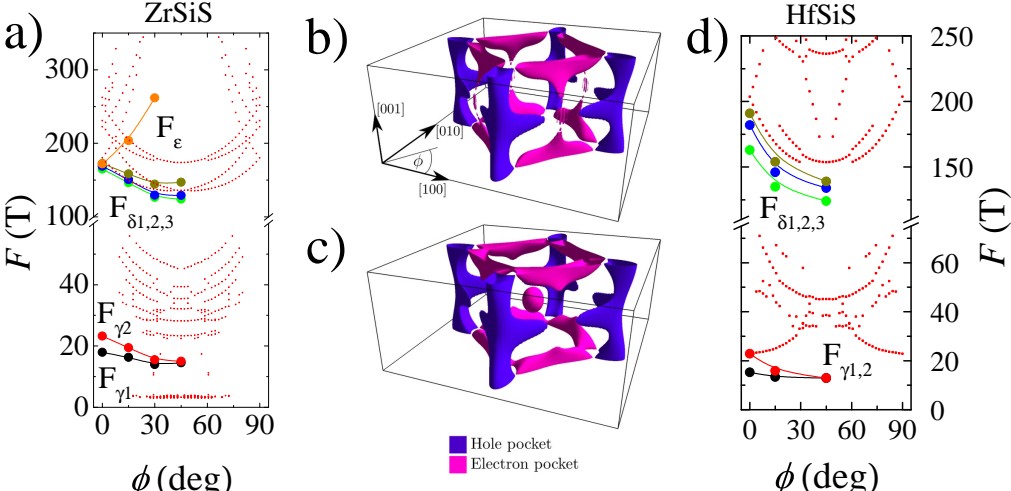

Figure 4: a) Comparison of experimentally detected and DFT calculated frequencies in azimuth direction ($\phi$) for ZrSiS. There is a decent matching for the set of $F_{\delta 1,2,3}$ and $F_\epsilon$ frequencies. Whereas the DFT completely misses the low-frequency set. b) and c) calculated shape of the Fermi surface. With margenta are drawn the hole pockets and with blue the electron pockets. A change from Zr to Hf mainly affects Fermi surface states around $k_z = 0$ plane. d) Comparison of experimental and DFT calculated frequencies for HfSiS. We see similar results as in ZrSiS, only decent agreements is achieved for $F_\delta$ set of frequencies.

frequency slightly deviates from the DFT predictions but still shows a very strong dispersion and could be attributed to an elongated electron pocket located at the edge of the Brillouin zone ($k_z = \pm\pi/2c$ plane). $F_{\delta 1,2,3}$ could also be associated with the same pocket. On the other hand, the observed low-frequency data set is not well reproduced by the calculations. Thus, it is very difficult to deduce which part of the Fermi surface would hosts these orbitals. The same conclusion could be drawn for HfSiS (Fig.4 d)).

Fig.4 b) and c) show calculated Fermi shapes for ZrSiS and HfSiS, respectively. Both Fermi surfaces have similarities regarding the shape of the hole pockets (red). Looking at the electron pockets (purple) we see that in ZrSiS upper and lower electron pockets are connected, whereas in HfSiS these pockets are well separated with additional pocket at the center of the Brillouin zone. The present DFT calculations have not revealed clearly observed low-frequency oscillations. Thus, further refinement of the ab-initio calculations (lattice parameters and Fermi energy position) is needed to identify the location of the low-frequency orbitals.

## 4 Conclusion

We have conducted detailed in-plane measurements of dHvA quantum oscillations in ZrSiS and HfSiS single crystals and compared the obtained data with the preformed ab-initio DFT calculations. The FFT analysis of the oscillations revealed a similar behavior in both iso-structural compounds. In the low-frequency range two distinct frequencies, $F_{\gamma 1}$ and $F_{\gamma 2}$, can be identified. At $\phi = 0^o$ ([100] direction) $\gamma 1$ and $\gamma 2$ are well separated, whereas at $\phi = 45^o$ ([110] direction) these two frequencies merged together. This indicates that $F_{\gamma 1}$ and $F_{\gamma 2}$ come from the same part of the Fermi surface which can facilitate small electron orbitals. The present DFT calculations do not reveal any low-frequency oscillations that were observed by experiments.

Further refinement of the ab-initio calculations is required to identify the location of the low-frequency orbitals. Whereas in the high-frequency region located between 100 T and 200 T there is a better agreement between the experiment and the DFT calculations allowing us to associate these frequencies with the electron pocket located around $k_z = \pm\pi/2c$ edge-planes of the 1st Brillouin zone.

## Acknowledgements

This work was supported by the CSF under the project IP 2018 01 8912 and CeNIKS project co-financed by the Croatian Government and the EU through the European Regional Development Fund-Competitiveness and Cohesion Operational Program (Grant No. KK.01.1.1.02.0013).

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
