# Peer review of "In-plane Fermi surface mapping of ZrSiS and HfSiS by de Haas-van Alphen oscillations"

_SciPost Physics Proceedings, doi:SciPost Phys. Proc. 11, 019 (2023)_

## Round 1 · Referee Report · Anonymous (Referee 1) · 2022-9-22

Report

The authors present a quantum oscillation study for B//x to B//y on ZrSiS and HfSiS. The extracted quantum oscillation frequencies are compared to their DFT calculations, and a moderate agreement is found for a selection of the observed frequencies. For one pocket, the authors extract a small effective mass, as expected for a nodal-line semimetal. This pocket is not reproduced by their DFT calculations.

The authors thus present a nice combination of experiment and theory. However, there are a number of points that the authors should address before the manuscript can be suitable for publication.

Requested changes

1. In the introduction, the authors introduce the materials ZrSiS and HfSiS. The paragraph that focuses on ZrSiS contains many citations. The following paragraph concerns HfSiS, and this paragraph does not contain any references, although there are relevant papers that concern this material. The authors should include these. Examples are for B//c Van Delft et al. (DOI 10.1103/PhysRevLett.121.256602), and Kumar et al. (DOI 10.1103/PhysRevB.95.121109) for B//x and B//c.

2. The authors present the Lifshitz-Kosevich formula in its full form, but only refer back to the mass or temperature-dependent part later on. First, it would be good if they only describe what is relevant for their analysis.
On a textual note: the authors write “$\omega_c$ = eB/m$_c$ is cyclotron mass”, and I think they intend to write that this is the cyclotron frequency, rather than cyclotron mass. In addition, they could also extract the Dingle temperature.

Regarding the analysis of quantum oscillations
3. The authors measure frequencies $F_{\gamma_{1,2}}$ and $F_{\delta_{1,2,3}}$. How do their measurements and extracted frequencies compare to what is reported in literature, for example by Kumar et al.? It would be insightful if they comment on how their measurements fit in the literature.

4. The authors identify the orbits $F_{\gamma_{1,2}}$ and $F_{\delta_{1,2,3}}$. In previously published papers, orbits at close frequencies are called differently. Are these the same frequencies? If they are, it would be good to have consistent terminology.

5. The FFT in Figure 3d) shows peaks at slightly lower and higher frequencies than $F_{\delta_{1,2,3}}$: at ~170T and at ~210 T. What is the interpretation of these peaks of the authors? Are these real peaks, artefacts of the data analysis, or something else?

6. The authors have measured the quantum oscillations at many different temperatures at various angles. For $\phi$=45$^{\circ}$ they have extracted the effective mass on the $F_{\gamma_{1,2}}$ pocket. It would be valuable to extract the masses of the remaining frequencies too, and I think that this is possible with the temperature dependence that the authors have measured.

7. Regarding the extracted effective mass of the $F_{\gamma_{1,2}}$ frequency in both HfSiS and ZrSiS: what is the error bar of these effective masses? Could the authors add these to their plots (fig 2d, 3c) and text?

8. On page 5, the authors write:
“Due to the high sample quality, the oscillations can be detected even at 1 T.”
Whether quantum oscillations can be observed also depends on the effective masses. A correct statement would be:
“Due to the high sample quality and low effective masses, the oscillations can be detected even at 1 T.”

Regarding the DFT calculations
9. These DFT calculations can be more valuable, if the authors put the extract frequencies for specific field angles in a table and compare theory to experiment. Can the authors also extract the masses from their calculations, and compare to their experimentally set effective masses?

10. For readability, it would be good if the authors could indicate the orbits including names on Fig. 4b and c, e.g. for B//x.

11. I found multiple papers in literature with DFT calculations of HfSiS and ZrSiS. How are the current results placed in this context, how do they fit in, and what do they add? Can the authors discuss this?

12. The legend of the DFT figures (Fig 4bc) reads blue = hole pocket and magenta = electron pocket. In contrast, the authors write in the caption of the figure and the text that concerns this figure that red = hole pocket and purple = electrons pocket. This is different from each other. Can the authors use consistent language?

13. The authors could improve their story. In the introduction, they write about interesting features of nodal line semimetals and their masses. For their story, it would be good to return to their initial motivation at the end of the paper.

---

## Round 2 · Author Response

Dear Referee, Editor,
We are very grateful for your comments and time invested in reading our manuscript, and here we address them.
Some of the suggested questions stated by the referee require additional research and will be addressed in a future manuscript.
A list of corrections and reports to the referee's comments/suggestions has been given in a separated MS Word document.

---

## Round 2 · List of Changes

Page 1. Introduction
Two references have been added, referenced added [18] and [24]

Page 5. Results and discussion
“ωc = eB/mc is cyclotron mass" is replaced with "ωc= eB/mc is cyclotron frequency"
“Due to the high sample quality, the oscillations can be detected even at 1 T.” is replaced with "Due to the high sample quality and low effective masses, the oscillations can be detected even at 1 T.”

Fig 2. and Fig 3. The error bars for the effective mass have been added in the figure caption.

---

## Editorial Decision

published